

# The effects of shoe structural features on agility and stability tasks during walking

Kavya Katugam-Dechene,  Ava Cook,  Anh Nguyen,  Ross Smith,
Andrew Shelton and  Jason R. Franz

Lampe Joint Department of Biomedical Engineering, University of North Carolina at Chapel Hill and North Carolina State University, Chapel Hill, NC, United States of America

## ABSTRACT

**Background**. Footwear can accommodate foot pathologies, alleviate symptoms of musculoskeletal injury, provide environmental protection and, for orthopedic or aging consumers, enhance agility or stability to improve daily locomotion. Independent of the specific characteristics underlying footwear selection, shoes are often called upon to support the performance of many different types of activities.

**Purpose**. The purpose of this study was to investigate the extent to which shoe design features intended for stability versus agility affect walking tasks that would disproportionately depend on those features.

**Methods**. Fourteen adults completed two walking tasks intended to disproportionately require greater agility or greater stability. Participants completed walking tasks while wearing each of two footwear designs: a supportive hiking boot and a flexible sneaker.

**Results**. We found no significant performance differences between footwear designs in either the agility task metrics or stability task metrics. Conversely, participant perceptions reflected differences in footwear design features.

**Conclusions**. The results of this study suggest that shoe design features intended for stability versus agility minimally bias walking performance towards either respective benefit. Our results may improve consumer confidence in footwear selection, often thereafter called upon to meet the needs of a variety of activities of daily living.

## INTRODUCTION

Footwear is the interface between the body and the ground surface on which we walk, influencing load transmission, sensory feedback, and joint kinematics. As such, a pivotal and unavoidable relation exists between shoe structural features, agility, and stability during locomotion. Independent of the specific characteristics underlying footwear selection, shoes are often called upon to support the performance of many different types of everyday activities, from steady-state walking to turns and reactive balance tasks in our communities. Shoes are often marketed as being specialized for specific functions due to their unique combination of design characteristics, *e.g.*, stability shoes for individuals with balance impairments or at greater risk of falling, or agility-focused features for those who are more active. However, little is known about the generalizability of specialized shoe designs across activities with varying performance demands.

Corresponding author
Jason R. Franz, jrfranz@email.unc.edu

As the first focus area of this manuscript, specific footwear design characteristics can affect stability as a measure of locomotor performance. For example, design features such as ankle collar height, heel closure design, shoe width, sole firmness, and midsole thickness have been shown to influence balance control and stability (*Edelstein, 1987*; *Lord et al., 1999*; *Menz, Morris & Lord, 2006*; *Menant et al., 2008*; *Aboutorabi et al., 2016*; *Menz, Auhl & Munteanu, 2016*). Thus, selection of appropriate footwear can be important in augmenting stability during locomotor tasks.

As the second focus area of this manuscript, also important to locomotor performance and potentially footwear selection is agility—the ability to quickly initiate movements, change directions, and/or change speed (*Sheppard & Young, 2006*). Inherently, the design and structure of footwear, as the interface between the body and the ground, can influence the user's ability to successfully navigate agility-dependent tasks. Both shoe sole stiffness and outsole traction have been shown to influence user performance during athletic movements such as cutting and vertical jumping (*Stefanyshyn & Nigg, 2000*; *Tinoco, Bourgit & Morin, 2010*; *Worobets & Wannop, 2015*; *Alirezaei Noghondar & Bressel, 2017*). These data support the notion that selection of appropriate footwear for a given movement task can be important in ensuring agility.

The use of "inappropriate" footwear, *e.g.*, slippers, backless shoes, or high heels (*Lord et al., 1999*; *Menz, Morris & Lord, 2006*), is indicated as a risk factor for falls in several populations, including older adults and hospital workers (*Gabell, Simons & Nayak, 1985*; *World Health Organization, 2021b*). Intuitively, footwear geared towards an aging demographic and/or orthopedic consumers may opt to consider characteristics designed to enhance walking agility and/or stability to improve locomotor performance during activities of daily living (*McPoil, 1988*; *Maki & McIlroy, 1997*; *Koepsell et al., 2004*). However, it remains possible that specialized footwear designs disproportionately improve performance during target activities, while simultaneously hindering performance in others. For example, large, "bulky" shoes with stiffer outsoles designed to aid stability could hinder user agility. Despite extensive research on footwear and postural control, few studies have directly tested how stability-enhancing features affect dynamic agility tasks, or whether agility-oriented designs compromise balance under perturbation. The notion of an agility-stability tradeoff during posture and locomotion is well established (*Hasan, 2005*; *Jindrich & Qiao, 2009*; *Huang & Ahmed, 2011*; *Ting et al., 2015*; *Acasio et al., 2017*), though few if any studies have aimed to quantify this tradeoff with regard to the interaction with footwear.

The purpose of this study was to investigate the extent to which shoe design features intended for stability *versus* agility affect walking tasks that would disproportionately depend on those features. Compared to those in the opposing shoe, we hypothesized that: (1) shoe features intended for stability would decrease the vulnerability to treadmill-induced slip perturbations, while (2) shoe features intended for agility would improve performance on a figure-8 walking task. Data in support of these hypotheses would indicate differences in agility and stability metrics dependent on the shoe design being worn, justifying the need for specialized, task-specific footwear features. Alternatively, rejecting these hypotheses

Table 1 **Participant demographics ($n = 14$).** Values are reported as group mean $\pm$ group standard deviation.

| Demographic outcome | |
|---|---|
| Sex | 7 males, 7 females |
| Age | $45 \pm 20$ years old (range: 24 to 75 years old) |
| Height | $1.71 \pm 0.10$ m |
| Mass | $69.6 \pm 11.0$ kg |
| Preferred walking speed | $1.32 \pm 0.20$ m/s |

would instead indicate that a generalized shoe design is sufficient to meet the needs of a variety of daily living activities.

## MATERIALS & METHODS

We conducted a laboratory-based, within-subjects experimental study using a repeated-measures design to evaluate how two consumer-relevant footwear types, one designed for stability and one for agility, affect locomotor performance and perceived performance during walking tasks that disproportionally challenge agility or stability.

### Participants

Fourteen healthy adults (seven female, age: $55.1 \pm 22.1$ yrs., height: $1.61 \pm 0.05$ m, mass: $63.9 \pm 7.7$ kg; seven male, age: $29.3 \pm 5.8$ yrs., height: $1.80 \pm 0.05$ m, mass: $74.5 \pm 12.8$ kg) participated in this study. Participant demographics are summarized in Table 1.

An *a priori* power analysis indicated that a sample size of $n = 14$ would provide 90% power to detect a minimum effect size (Cohen's d) of 0.81 at an alpha level of 0.05 in a within-subjects design. Following exclusions due to marker occlusions, the final sample size ($n = 12$) retained sufficient power to detect a slightly larger minimum effect size, with the threshold increasing to Cohen's d = 0.89. Additional information can be found in Article S2.

We included participants over 18 years old with no recent bone breaks or injury requiring surgical intervention (*i.e.,* 6 months), no use of prostheses, without neurological, musculoskeletal, or cardiopulmonary disease, and who could walk without the use of an assistive device.

In collaboration with the study sponsor, we intentionally developed an enrollment strategy to reflect the diversity of the orthopaedic consumer population, which varies widely in age, sex, demographic background, and underlying conditions. Although this approach is somewhat unconventional for a study of this scale, it was designed to enhance the relevance and real-world applicability of our findings. We recruited subjects *via* flyer, email, and word of mouth. Subjects that showed interest in participating were directed towards an online screening survey. Eligible participants were then invited to participate in the remainder of the study while ineligible participants were informed of their results. Participation in this study comprised of a single laboratory visit to the Applied Biomechanics Lab in the Lampe Joint Department of Biomedical Engineering at the University of North Carolina at Chapel Hill. The study protocol was approved by the University of North

(A) Stability-Focused Design     (B) Agility-Focused Design

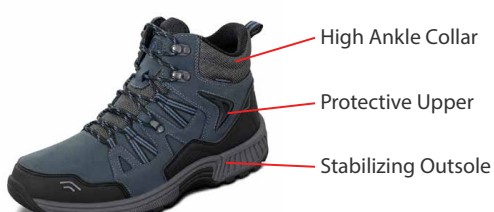
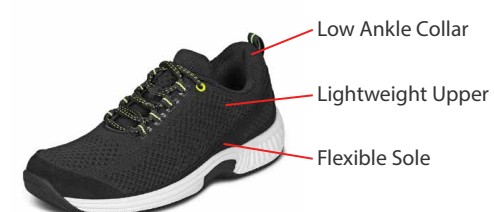

High Ankle Collar     Low Ankle Collar

Protective Upper     Lightweight Upper

Stabilizing Outsole     Flexible Sole

**Figure 1  Two footwear designs worn by participants.** (A) The hiking boot (Orthofeet© Women's Dakota Boot or Orthofeet© Men's Ridgewood Boot) was designed to improve stability during movement, including features such as a thick upper material to protect the foot, a high ankle collar to decrease potential for excess ankle mobility, and a stabilizing outsole design. Photo credit: Orthofeet, Inc. (B) The sneaker (Orthofeet© Women's Coral Sneaker or Orthofeet© Men's Lava Sneaker) was designed to allow for agility and maneuverability, including features such as a flexible sole, lightweight and flexible upper material, and a low ankle collar, all to reduce impedance to movement. Photo credit: Orthofeet, Inc.

Carolina Chapel Hill Biomedical Sciences Institutional Review Board (IRB Number: 22-2295), and all subjects provided written informed consent prior to participating.

## Equipment

A 16-camera motion capture system (Motion Analysis Corporation, Santa Rosa, CA, USA) recorded kinematic data at 100 Hz. Participants walked on a dual-belt, instrumented treadmill (Bertec, Columbus, OH, USA). We fitted participants with 40 retroreflective markers placed as follows: 26 markers on the anterior and posterior iliac spines, medial and lateral femoral condyles, medial and lateral malleoli, posterior calcanei, 1st and 5th metatarsal heads, acromials, sacrum, 7th cervical spine, 10th thoracic spine, sternum, sternal notch, an arbitrary location on the right scapula for asymmetry; and an additional 14 tracking markers placed in clusters on the lateral thighs and shanks. These data were collected as part of a larger study on aging, gait, and balance, and only the posterior sacrum and distal leg markers were used for analyses in this study.

## Footwear

The shoes used in this study were provided by Orthofeet, Inc., a company that designs footwear, insoles, and socks for consumers with mobility issues and/or foot conditions. We focused our experimental comparisons in this study on two footwear types: a supportive hiking boot and a flexible sneaker-type shoe (Fig. 1). Mechanical characteristics for each footwear model used in this study are summarized in Table 2. For each of the footwear types, we measured heel height and heel-to-toe drop using digital calipers. We measured the stiffness of the insole and outsole using standard durometers, reported in Shore O and Shore A units, respectively.

First, we selected a hiking boot with a more supportive midsole, a stiffer outsole, a higher ankle collar, and a lesser heel-to-toe drop (Orthofeet© Women's Dakota Boot or Orthofeet© Men's Ridgewood Boot; Fig. 1A). These characteristics are intended to
**Table 2 Mechanical properties of the footwear used in the study.** Values are shown for heel height, heel-to-toe drop, and insole and outsole stiffness for the "agility" and "stability" shoes, respectively.

|  | "Agility" Shoe | "Stability" Shoe |
|---|---|---|
| Heel height | 24.4 mm | 25.6 mm |
| Heel-to-toe drop | 14.1 mm | 9.8 mm |
| Insole stiffness | 30 Shore O | 30 Shore O |
| Outsole stiffness | 40 Shore A | 70 Shore A |

promote stability by ensuring a stable center of pressure progression, enhancing traction to prevent slipping, and reducing ankle motion, respectively.

Second, we selected a sneaker with lightweight construction, less stiff outsole, a lower ankle collar, and a greater heel-to-toe drop (Orthofeet© Women's Coral Sneaker or Orthofeet© Men's Lava Sneaker; Fig. 1B). These features are presumed to promote agility *via* reducing energy requirement for foot progression, supporting natural foot motion, and increasing ankle mobility, respectively.

There is great value in investigating the isolated effects of distinct shoe features that may be intended to promote locomotor agility and/or stability. However, from a consumer perspective, footwear designs rarely differ in singular modifications to their construction and often benefit collectively from many design features. This is most certainly the case for the shoes tested and compared in this study, with robust consumer-relevant differences not isolated to single design features, thereby mimicking the actual product landscape. Thus, we view the extent of these differences to be a strength and not a limitation of our study.

## Procedures

We first computed the preferred walking speed (PWS) for each participant as the average of four 30-meter overground walking trials timed using photocells (Bower Timing Systems, Draper, UT, USA). Participants then completed a three-minute warm-up walk at their PWS before data collection to acclimate to treadmill walking.

Participants completed two walking tasks in each shoe type, with shoe conditions chosen in block-randomized order. First, participants completed a task that we considered representative of demands on locomotor agility. Specifically, participants performed a "figure 8" walking task (Fig. 2) five times with instruction to walk at a "comfortable" pace, starting from and ending at still-standing, navigating around two cones placed 5 ft apart, in a figure 8 motion. Second, participants completed a task known to precipitate locomotor instability (*Crenshaw & Grabiner, 2014*; *Lee, Bhatt & Pai, 2016*; *Liu, Bhatt & Pai, 2016*; *Shelton et al., 2024*). Specifically, participants walked on a dual-belt, instrumented treadmill at their PWS while responding to a series of treadmill-induced slip perturbations (Fig. 3). Participants experienced sudden treadmill belt decelerations at 6 m/s² lasting 200 ms in duration, applied to five randomly-selected heel strikes per leg. Each deceleration event was separated by at least 10 steps. Following each 200 ms perturbation, the treadmill belt returned to the preferred walking speed at six m/s². A MATLAB script (MathWorks, Natick, MA, USA) used in previously-published studies controlled the treadmill belts during the

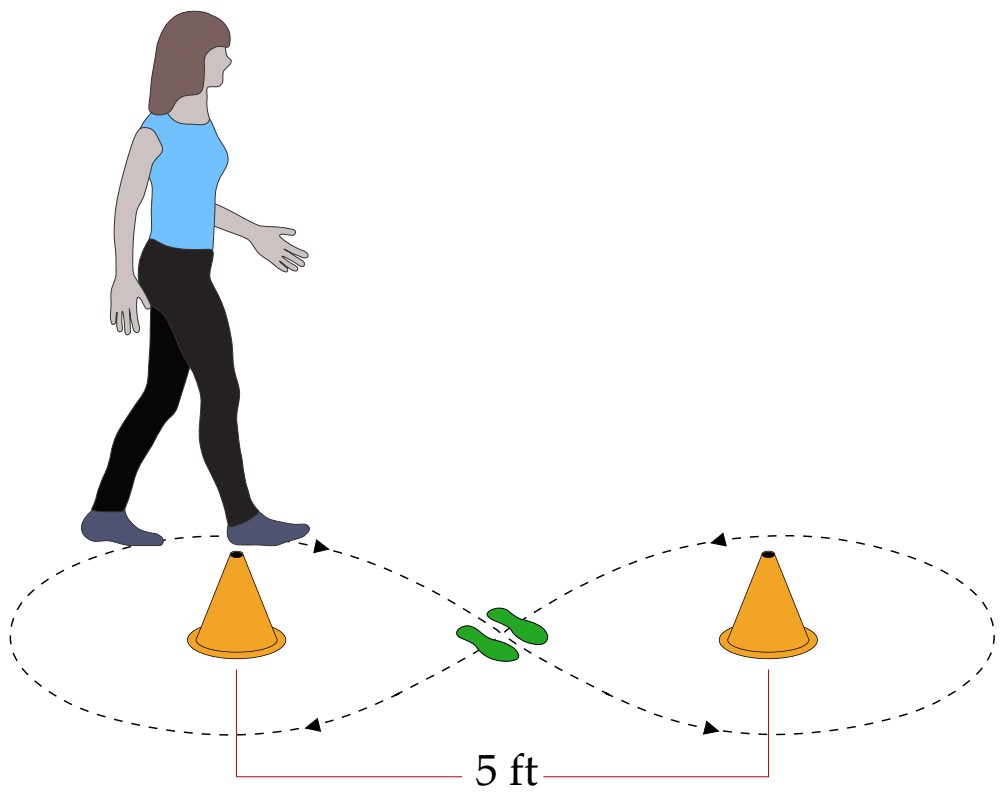

**5 ft**

**Figure 2** **Figure-8 walking task.** The figure-8 walking task involved both turning and straight walking and was used to assess performance during agility-intensive movements. Participants started from still standing, positioned between two cones placed 5 ft apart. They were then instructed to walk at a "comfortable" pace, navigating a figure-8 pattern around the two cones before coming to a full stop at the starting point. Each participant repeated this task five times in each shoe condition.

stability task (*Crenshaw & Grabiner, 2014*; *Lee, Bhatt & Pai, 2016*; *Liu, Bhatt & Pai, 2016*; *Shelton et al., 2024*).

It has been shown that specific footwear design characteristics can affect both user performance (*Edelstein, 1987*; *Lord et al., 1999*; *Stefanyshyn & Nigg, 2000*; *Menz, Morris & Lord, 2006*; *Menant et al., 2008*; *Tinoco, Bourgit & Morin, 2010*; *Worobets & Wannop, 2015*; *Aboutorabi et al., 2016*; *Menz, Auhl & Munteanu, 2016*; *Alirezaei Noghondar & Bressel, 2017*) and user perception (*Amiez et al., 2021*). Further, the interaction between objective locomotor performance and self-perceptions is well-established (*Luo et al., 2009*; *Li & Huang, 2022*; *Shelton et al., 2024*). Thus, we also measured participants' perceptions of their performance during each task in each footwear condition using a custom questionnaire (Article S1).

## Data analysis

Primary outcome measurements from agility trials were average cycle completion speed, length along the major longitudinal axis, width along the minor transverse axis, and steps per cycle. Primary outcome measurements from stability trials were step width,

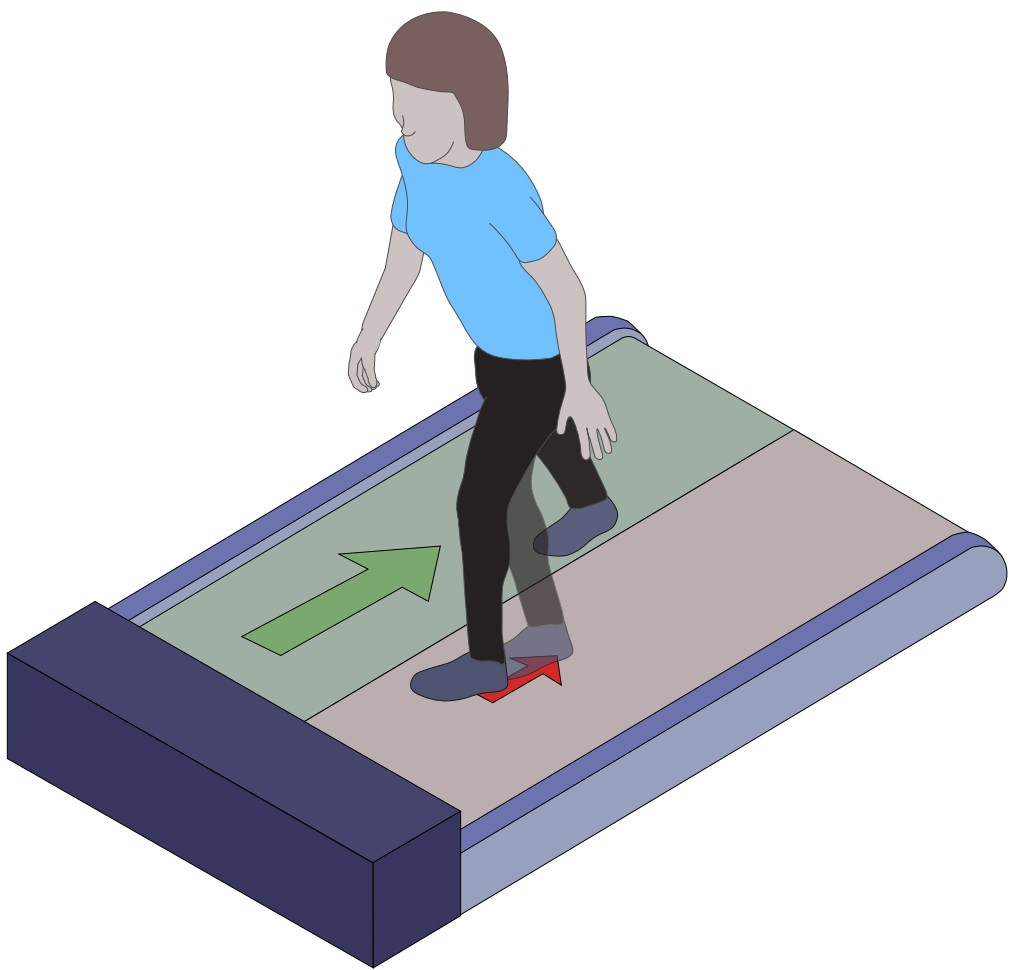

**Figure 3** **Treadmill-induced slip perturbation walking task.** The stability-intensive walking task consisted of walking while responding to unanticipated treadmill-induced slip perturbations. Participants walked on a dual-belt, instrumented treadmill at their preferred walking speed. Slip perturbations were applied to five randomly-selected heel strikes per leg and were induced *via* sudden treadmill belt decelerations (6 m/s², 200 ms in duration). Each deceleration event was separated by at least 10 steps. Following each 200 ms perturbation, the treadmill belt returned to the preferred walking speed at an acceleration of 6 m/s².

step length, and margin of stability (MoS) both in the anteroposterior ($\text{MoS}_{\text{AP}}$) and the mediolateral ($\text{MoS}_{\text{ML}}$) directions. We analyzed marker data as previously described by our group (*Shelton, 2024*; *Shelton et al., 2024*). We calculated MoS using published guidelines (*Hof, Gazendam & Sinke, 2005*; *McAndrew Young, Wilken & Dingwell, 2012*; *Richards et al., 2019*). First, we filtered marker position data using a 4th order low-pass Butterworth filter with a cutoff frequency of 12 Hz. We calculated the approximate location for the center of mass ($x$) as the center of the polygon created by the anterior and posterior iliac spine marker positions (*Rosenblatt & Grabiner, 2010*; *Hak et al., 2013*; *Peebles et al., 2017*; *de Jong*

*et al., 2020*). We next calculated extrapolated center of mass (xCoM) using:

$$xCoM = x + \frac{\dot{x} + V_{treadmill}}{\omega_0}$$

where $x$ is the CoM position, $\dot{x}$ is the CoM velocity, $V_{treadmill}$ is the treadmill belt velocity, and $\omega_0$ is the natural frequency of an inverted pendulum model of the stance phase. We calculated $\omega_0$ as:

$$\omega_0^2 = \frac{g}{L}$$

where $g$ is acceleration due to gravity and $L$ is participant leg length. We estimated leg length as the mean distance between the sacral marker and the heel marker at heel-strike. We calculated MoS as the distance between the boundary of the base of support (BoS), defined as the 1st metatarsal marker for anteroposterior MoS or as the 5th metatarsal marker for mediolateral MoS, and the xCoM projected to the treadmill belt.

We extracted the MoS at heel strike of the left and right legs. For the normal walking trials, mediolateral ($MoS_{ML}$) and anteroposterior ($MoS_{AP}$) MoS were averaged across all strides. MoS outcomes were calculated at the instant of heel strike directly following perturbation onset (*i.e.,* the recovery step) and then averaged across all perturbation occurrences within the trial (*Martelli et al., 2016*; *Golyski et al., 2022*) following published guidelines for discrete perturbations.

## Statistical analysis

Statistical analyses were conducted using R (*R Core Team, 2024*). One-way analysis of variance (ANOVA) was used to assess differences between footwear conditions for primary outcome measures from agility trials (steps per cycle, average cycle completion speed, length along the major longitudinal axis, and width along the minor transverse axis) and stability trials (step width, step length, $MoS_{AP}$, and $MoS_{ML}$). Prior to conducting ANOVA, the normality of residuals was evaluated using the Shapiro–Wilk test, and homogeneity of variances was assessed using Levene's test. If assumptions were violated, a non-parametric Kruskal-Wallis test was applied instead. Effect size (ES) between footwear conditions was computed as Cohen's d, where values of 0.2, 0.5, and 0.8 indicated small, medium, and large effects, respectively. An alpha level of 0.05 was used for all tests.

## RESULTS

Primary outcome measures categorized by footwear condition for the fourteen adult participants are summarized in Tables 3 (agility task) and 4 (stability task). These results are also shown in Fig. 4 (agility task) and Fig. 5 (stability task).

Data on primary outcome measures from the agility task are reported for all 14 subjects (Table 3 and Fig. 4). All primary outcome measures from the agility task (steps per cycle, average cycle completion speed, length along the major longitudinal axis, and width along the minor transverse axis) met the assumption of normality, as assessed by Shapiro–Wilk tests ($p > 0.05$ for all variables). We found no significant differences between footwear designs in average number of steps taken, average figure-8 speed, minor axis width, or
**Table 3 Primary outcome measures for the agility task (figure-8, $n = 14$).** Values are reported as group mean ± group standard deviation. Effect size (ES) computed as Cohen's d.

|  | "Agility" Shoe | "Stability" Shoe | P | ES |
|---|---|---|---|---|
| Step count | 14.8 ± 2.3 | 15.0 ± 2.0 | 0.45 | 0.09 |
| Walking speed (m/s) | 0.79 ± 0.09 | 0.78 ± 0.08 | 0.27 | 0.12 |
| Minor axis width (m) | 1.12 ± 0.22 | 1.12 ± 0.22 | 0.78 | 0.03 |
| Major axis length (m) | 2.50 ± 0.35 | 2.53 ± 0.34 | 0.51 | 0.07 |

**Table 4 Primary outcome measures for the stability task (treadmill-induced slip perturbations, $n = 12$).** Values are reported as group mean ± group standard deviation. Effect size (ES) computed as Cohen's d.

|  | "Agility" Shoe | "Stability" Shoe | P | ES |
|---|---|---|---|---|
| Step width (m) | 0.17 ± 0.03 | 0.17 ± 0.04 | 0.43 | 0.12 |
| Step length (m) | 0.69 ± 0.08 | 0.66 ± 0.15 | 0.33 | 0.27 |
| $MoS_{AP}$ (cm) at heel strike | −0.48 ± 10.62 | −1.80 ± 11.32 | 0.51 | 0.12 |
| $MoS_{ML}$ (cm) at heel strike | 15.65 ± 1.86 | 16.01 ± 1.66 | 0.38 | 0.20 |

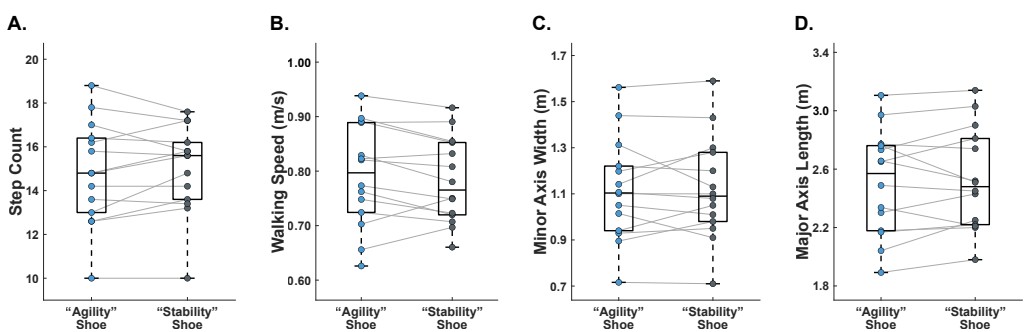

**Figure 4 Primary outcome measures for the agility task (figure-8, $n = 14$).** Each data point indicates the average performance of 5 runs. Grey lines connect data between footwear conditions within a given participant. (A) Steps per cycle. (B) Walking speed (m/s). (C) Length along the major longitudinal axis (m). (D) Width along the minor transverse axis (m).

major axis length. Six subjects took more steps when wearing the sneaker, six subjects took more steps when wearing the hiking boot, and two subjects had an equal average step count between footwear conditions. When wearing the sneaker, nine subjects walked faster, eight subjects had a greater minor axis width, and five subjects had a greater major axis length.

Data on primary outcome measures from the stability task, due to marker occlusions, are reported for 12 of the 14 subjects (Table 4 and Fig. 5). All primary outcome measures from the stability task (step width, step length, $MoS_{AP}$, and $MoS_{ML}$) met the assumption of normality, as assessed by Shapiro–Wilk tests ($p > 0.05$ for all variables). We found no significant differences between footwear designs for step width, step length, $MoS_{AP}$, or $MoS_{ML}$. When wearing the sneaker, four subjects took wider steps, seven subjects took longer steps, eight subjects had a greater $MoS_{AP}$, and five subjects had a greater $MoS_{ML}$.

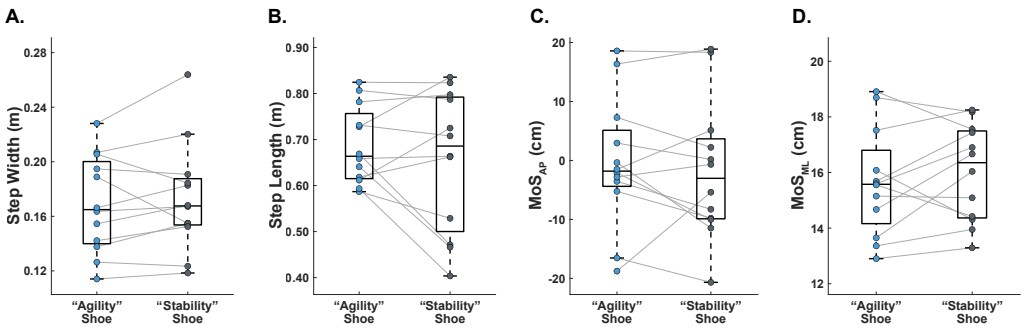

**Figure 5 Primary outcome measures for the stability task (treadmill-induced slip perturbations, $n =$ 12).** Grey lines connect data between footwear conditions within a given participant. (A) Step width (m). (B) Step length (m). (C) $MoS_{AP}$ (cm). (D) $MoS_{ML}$ (cm).

## DISCUSSION

Contrary to our hypothesis, the results of this study suggest that shoe design features intended for stability *versus* agility minimally influence subject performance when faced with various walking challenges. These minimal differences in performance may reflect the relatively modest differences in mechanical properties between the two footwear conditions. For instance, while the outsole stiffness of the stability shoe was substantially higher (70 Shore A *vs.* 40 Shore A), other mechanical differences, such as heel height and heel-to-toe drop, were relatively small and may not have been sufficient to elicit marked performance changes during the walking tasks evaluated. More specifically, design features intended for agility or stability seemed to convey generalized performance characteristics on walking tasks we would presume to disproportionately depend on those features. As we discuss in more detail below, we suggest that our results may improve consumer confidence in footwear selection—often thereafter called upon to meet the needs of a variety of activities of daily living.

Differences in footwear design features minimally influenced objective subject performance—at least that characterized by our selection of outcomes and specific tasks. Conversely, subjects expressed differences in perception of performance that were task-specific and reflected differences in the design features of these two shoes. Perceptions of the hiking boot were mixed: some felt the shoe to be supportive, making them perceive increased stability in their movements. However, others disliked the design features intended for support, such as the "bulky" protective upper material or the high ankle collar design which serve as prerequisites for ankle support. These design features, along with the increased outsole stiffness (70 Shore A), may have contributed to the perception of rigidity and reduced freedom of movement. These latter participants reported the impedance to movement more noticeable than the structural benefits to stability. Conversely, for the sneaker, participants reflected positively on the lightweight design and flexibility, perceiving a greater ability to quickly react to and correct for perturbations. Further, only one subject preferred the hiking boot over the sneaker, with most subjects preferring the "maneuverability" afforded by the sneaker over the "stability" afforded by the hiking boot.
In summary thus far, despite a clear dichotomy of perception between the two shoe designs, there were no significant differences in metrics of agility or stability between shoe conditions. Studies have shown that specific footwear design characteristics can affect user perception of stability and balance control. *Amiez et al. (2021)* found that specifically designed balance shoes improved user-perceived stability and safety compared to personal footwear. The consistency between our findings and those of *Amiez et al. (2021)*, suggests that perceptual and mechanical features of footwear—such as outsole width and stability-enhancing design—may influence not just objective measures of sway, but also perceived safety and balance confidence. This dual effect has important implications for adherence to fall-prevention footwear interventions in community settings.

A study by *Park et al. (2017)* found that increased sole stiffness significantly increased user comfort but not performance or lower extremity kinematics during an agility-dependent task. These findings parallel ours, showing that shoe characteristics intended to influence stability and/or agility, such as differences in outsole stiffness, may instead have greater impact on both comfort and perception of performance. Our shoes differed most substantially in outsole stiffness, which likely influenced perception more than actual performance, supporting *Park et al.*'s (*2017*) conclusion that increased stiffness can alter subjective comfort without measurable biomechanical changes. These findings imply that when optimizing outsole stiffness in performance footwear, designers should consider user-specific comfort thresholds, as perceived comfort may indirectly influence movement quality, fatigue, or willingness to wear the shoe during extended play.

In a cohort of older adults, *Azhar, Munteanu & Menz (2023)* found that differences in footwear design (*i.e.,* minimalistic *vs.* supportive) did not affect balance performance or walking stability, but did affect perceived stability and reported comfort. Like *Azhar, Munteanu & Menz (2023)*, we observed no significant difference in balance metrics across footwear conditions, suggesting that postural stability in older adults may be more resilient to modifications in footwear design than previously assumed. This may indicate a reliance on compensatory sensorimotor strategies that override subtle footwear-induced perturbations. These shared results also may suggest that for older adults without significant mobility impairments, footwear choice can prioritize comfort and aesthetics without compromising balance—a consideration that may improve long-term adherence to fall-prevention footwear recommendations.

In a study on footwear and balance, *Burke (2012)* found a clear association between perceived footwear comfort and neuromuscular control of balance in both healthy adults and adults with musculoskeletal disorders. Another study found that in older adults, life history, such as having experienced a fall, affects the perception of footwear comfort in relation to the control of balance (*Puszczalowska-Lizis et al., 2022*). These findings point to a complex interdependency between how users perceive footwear (*i.e.,* perception of comfort) and their performance in said footwear.

We suggest that when selecting shoes, user perception and comfort should be a greater priority. Psychological factors such as balance confidence and self-efficacy, particularly in older consumers, have been shown to influence inclination towards and participation in physical activity (*Talkowski et al., 2008*; *Cheval & Boisgontier, 2021*). Self-efficacy is defined

as the belief that one can successfully achieve a specific outcome (*Bandura, 1977*). Further, individuals with lower balance confidence show a greater prevalence of developing gait, balance, and cognitive disorders over time, ultimately resulting in decreased mobility (*Vellas et al., 1997*). Reduced physical activity has been identified as a behavioral risk factor for many health ailments across all ages, including but not limited to heart disease, type 2 diabetes, and some cancers (*Blair, 2009*; *World Health Organization, 2009*); bone loss (*Kos et al., 2014*), decreased motor control ability (*Canu et al., 2019*), weakened muscle properties (*Canu et al., 2019*), and increased falls risk (*World Health Organization, 2021b*; *World Health Organization, 2021a*). Thus, shoe design features that improve perceived ability may improve locomotor health by increasing engagement in physical activity. Importantly, even subtle mechanical differences, such as a higher heel-to-toe drop or stiffer outsole, may influence user preference and perceived competence, which in turn could shape long-term adherence to footwear use in real-world settings.

There are several limitations of this study. First and perhaps most influential, the two tasks we interpreted to disproportionately rely on the stability *versus* agility characteristics of footwear likely have significant overlap. For example, an agility-focused design may also convey important benefits for locomotor stability, while a stability-focused design may enhance confidence in performing agility tasks. Thus, it is difficult to isolate the two locomotor phenomena being evaluated. Additionally, the specific range of mechanical differences engineered into our footwear models may not have been sufficient to amplify distinct effects across tasks. Future work could systematically vary design parameters, such as stiffness, weight, and collar height, to better delineate the thresholds at which design influences performance. Given our findings at the interface of objective performance and perception, future experiments are warranted that continue to disassociate agility and stability requirements across a wider range of locomotor tasks.

In addition, it is possible that significant differences could be identified between footwear conditions in performance metrics using alternate outcome measures not included in this study. These may include numerous alternative balance and agility outcomes or measures of muscle excitation.

Lastly, our relatively small and heterogeneous sample may limit the generalizability of our findings. Orthopaedic consumers represent a diverse population, varying widely in age, demographic characteristics, underlying conditions, sex, and other factors. In close collaboration with our study sponsor, we intentionally designed the enrollment strategy to reflect this diversity. While we acknowledge that this approach is unconventional, we believe it enhances the relevance and applicability of our findings.

Future work could more precisely tease apart agility and stability demands, potentially by incorporating more diverse locomotor tasks, muscle activation metrics, or perturbation types. Additionally, investigating how specific shoe features (*e.g.*, sole stiffness, ankle collar height) individually contribute to perceived *versus* actual performance may inform more targeted footwear innovations. Longitudinal studies examining how changes in self-efficacy or comfort perception influence physical activity levels and fall risk, especially in older adults, could also extend the impact of this work into preventative and/or community health.

# CONCLUSIONS

The results of this study suggest that shoe design features intended for agility or stability offer minimal bias in walking performance towards either benefit, even on walking tasks presumed to depend on those specific features. These results have the potential to guide other footwear researchers to explore the interaction between subjective perception and objective locomotor performance across broader populations and locomotor task types.

We suggest that our results may enhance consumer confidence in their selection of footwear, which is thereafter called upon to meet the needs of a variety of activities of daily living. Moreover, psychological factors like balance confidence and self-efficacy, particularly in older consumers, can impact the inclination toward and participation in physical activities. Footwear designs that enhance confidence and perceived capability could potentially enhance locomotor health by encouraging greater engagement in physical activities.

# ACKNOWLEDGEMENTS

The authors would like to thank Shyam Patel for his assistance with data processing for this study.

## Funding

This research was supported by a grant from Orthofeet, Inc. to the Applied Biomechanics Laboratory at UNC Chapel Hill. The funders had no role in study design, data collection and analysis, decision to publish, or preparation of the manuscript.

## Grant Disclosures

The following grant information was disclosed by the authors:
Orthofeet, Inc. to the Applied Biomechanics Laboratory at UNC Chapel Hill.

## Competing Interests

The authors declare there are no competing interests.

## Author Contributions

- Kavya Katugam-Dechene analyzed the data, prepared figures and/or tables, authored or reviewed drafts of the article, and approved the final draft.
- Ava Cook performed the experiments, analyzed the data, authored or reviewed drafts of the article, and approved the final draft.
- Anh Nguyen performed the experiments, analyzed the data, authored or reviewed drafts of the article, and approved the final draft.
- Ross Smith conceived and designed the experiments, performed the experiments, analyzed the data, authored or reviewed drafts of the article, and approved the final draft.

- Andrew Shelton conceived and designed the experiments, performed the experiments, analyzed the data, authored or reviewed drafts of the article, and approved the final draft.
- Jason R. Franz conceived and designed the experiments, authored or reviewed drafts of the article, and approved the final draft.

### Human Ethics

The following information was supplied relating to ethical approvals (*i.e.*, approving body and any reference numbers):

The University of North Carolina Chapel Hill Biomedical Sciences Institutional Review Board (IRB Number: 22-2295).

### Data Availability

Raw data are available in the Supplemental Files.

### Supplemental Information

Supplemental information for this article can be found online at http://dx.doi.org/10.7717/peerj.19930#supplemental-information.

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
