# Peer review of "The effects of shoe structural features on agility and stability tasks during walking"

_PeerJ, doi:10.7717/peerj.19930_

## Round 0.1 · original submission · Major Revisions

The small and highly heterogeneous sample size, also pointed out by Reviewer 1, is a significant shortcoming in this study. as a consequence, the level of confidence in the conclusions that can be drawn from this study is low. The authors should carefully justify sample size and address the heterogenity. Additionally, the description of the statistical methods is inadequate. Key details, such as the specific tests performed, whether assumptions (e.g., normality of distribution) were checked, and the statistical software used for analysis, are missing.

Should you choose to revise the manuscript, please note that I disagree with Reviewer 2's recommendations concerning which papers should or should not be cited.

Reviewer 1 ·

Basic reporting

This study provides valuable insights into footwear sciences in terms of agility and stability. The manuscript is relatively well-written, the experiments are well-performed, and the results are well-presented. However, there are significant methodological concerns, as outlined below:

1. Heterogeneity of Subjects: The age range of the subjects (24-75 years) is excessively broad and inappropriate for this study, leading to unsound comparisons. It is recommended to narrow the age range, exclude elderly participants, and recruit more young participants. Alternatively, if the elderly population is of interest due to higher incidences of falls and deteriorated agility, this should be explicitly stated in the title, abstract, introduction, and discussion sections. Another option would be to divide the participants into different groups based on the age, which definitely requires a greater sample size.

2. Insufficient Sample Size: The number of participants (14) seems to be inadequate, and no power analysis has been provided. It is suggested to conduct an a priori power analysis using data from previous studies and recruit an appropriate number of participants accordingly.

3. Lack of Comprehensive Comparison: The discussion section lacks a thorough comparison with other studies. Additionally, the introduction is overly general and needs to be more specific regarding the study's background. These sections could be improved by citing and addressing more articles in footwear science. I believe that quite a few relevant studies can be found.

I am afraid I wouldn’t be able to address minor concerns until these major issues are resolved by the authors. Otherwise, please feel free to invite additional reviewers.

Apologize for brevity.

Experimental design

Maybe later

Validity of the findings

Maybe later

Additional comments

Maybe later

Reviewer 2 ·

Basic reporting

Clear and unambiguous, professional English used throughout.

Authors cut out outdated publications.

Experimental design

Research questions need to be defined.

The description of research methods requires supplementation.

The description of the results requires supplementation.

Validity of the findings

The results and conclusions require redrafting and additions.

Additional comments

This paper discusses an practical issue. The topic that has been conferred about in the paper is part of a comprehensive discussion on the problem of effects of shoe structural features on agility and stability tasks during walking. While I think this is an interesting topic, the manuscript could be improved.

Detailed comments are provided below:

Introduction
The introduction is based on citations of old publications. Please rewrite the introduction.

Material and Methods
The authors have to explain: when and where this study carried out? The work does not contain a sample size calculation and the graphic flow of the subjects through the study. The authors need to explain the procedure for recruiting study subjects, and provide the inclusion and exclusion criteria. The age range of the study subjects is too large. The age range and the mean with standard deviation should be given separately for females and males.
What test was used to check compliance with the normal distribution, whether the distribution of variables was normal?
There is no information regarding the consent of the Bioethics Committee.

Results
The variables need major explanation. The authors have to explain: All variables were consistent with the normal distribution? If the normality criteria are not met the authors should use the median, otherwise the can opt for the mean.

Discussion
The purpose of the work should not be given in the Discussion. The first sentence of the Discussion should be deleted. The Discussion is rather superficial and did not offer sufficient explanations for the mechanistic reasons behind the findings. Indeed, on occasions, although similarities with the literature were identified, no possible reason or implication was offered.

Conclusions
The section should be recast. Ultimately, it the current format, the conclusions offered simply do not provide significant guidance to clinicians or other researchers about what can be learned from this study nor does it address potential future research endeavors.

References
References such as references start with lines 304, 308, 310, 322, 326, 328, 337, 340, 344, 349, 365, 368, 371, 380, 386, 405, 407, 411, 417, 421 are too old, delete or update it.

General comments to the Authors
I have read through this paper several times and spent a fair amount of time reflecting on the meaning of this paper as well as the potential contribution it can make to the literature. The study design is not refined enough either. That said, my comments are offered with the intent of helping the authors improve this manuscript. When the authors address these issues I will be able to comment definitively and make the final decision.

---

## Round 0.2 · Major Revisions

One reviewer still has important comments.

Reviewer 1 ·

Basic reporting

I appreciate the authors' dedication to addressing the comments and enhancing the quality of the manuscript. Please find my comments for this round of review:

1. Thank you for providing further details on a priori sample size estimation and its justification. However, I still believe that what you received from your sample size estimation software should be considered within a homogeneous group. I still consider this a primary concern, as the generalizability and specificity of the findings remain in question.

2. In the conclusion section, authors need to state a brief description of the main findings and/or summarize the key points. Please move the suggestions for future studies and the potential impacts to the end of the discussion.

3. The descriptions of the footwear appear at the beginning of the Methods/Participants section which doesn’t seem to be a good place. Please relocate this information to the appropriate section, as you already have a dedicated Footwear section.

4. The descriptive statistics of the study participants must be in either the Methods/Participants section, or at the beginning of the Results section.

5. Sample size estimation must be mentioned in the Method/Participants section.

6. Please start your Methods section with the type and design of your study.

7. Please provide some information about the mechanical characteristics of the footwear used in your study (e.g., the stiffness [ShoreA/B] of the outsole and insole, upper layer materials, heel height, heel-to-toe-drop, and etc.). These characteristics can be measured or provided by the company. These must be also considered in your discussion when interpreting your findings and comparing the results.

8. You mentioned about collaboration with your study sponsor. I kindly ask you to provide a statement for conflict of interest as well as the funding resources. Also, given your statement ("For this study, we were intentional in designing our enrollment criteria to reflect the consumer segments identified by the study sponsor, rather than to represent the general population”) it appears that the study was conducted for the footwear providers (Orthofeet, Inc.). I would emphasize the importance of maintaining transparency throughout all aspects of the research to avoid any confusion and misunderstanding.

Best,

Experimental design

no comment

Validity of the findings

no comment

Reviewer 2 ·

Basic reporting

no comment

Experimental design

no comment

Validity of the findings

no comment

Additional comments

The authors responded to the reviewer's comments.

---

## Round 0.3 · accepted · Accept

I have reviewerd the final manuscript and the author responses to the latest reviewer comments. I'm satisfied with both and believe the manuscript is suitable for publication.